# Assessing early detection ability through spatial arrangements in environmental surveillance for poliovirus: A simulation-based study

Toshiaki R. Asakura[1,2,3]*, Kathleen M. O'Reilly[1,2]

1 Department of Infectious Disease Epidemiology and Dynamics, London School of Hygiene & Tropical Medicine, London, United Kingdom, 2 Centre for Mathematical Modelling of Infectious Diseases, London School of Hygiene & Tropical Medicine, London, United Kingdom, 3 School of Tropical Medicine and Global Health, Nagasaki University, Nagasaki, Japan

* toshiaki.asakura1@lshtm.ac.uk

## Abstract

Detecting the circulation of poliovirus in its early stages is paramount for swift public health action. While environmental surveillance (ES) is promising for enhancing early pathogen detection, the influence of spatial arrangement of ES sites on early detection remains unclear. Here, we aim to assess the early detection ability of ES by varying the number and location of ES sites using the simulation-based approach utilising geographic and demographic characteristics of South Africa as a case study of a non-endemic country. We developed a stochastic meta-population model among unimmunised children aged under 5 years old, assuming a single introduction of wild poliovirus serotype 1. We constructed six scenarios by combining three importation risk distributions (predicated on population size, approximations of international inbound travel volume and border crossing volume) with two ES site layout strategies (proportionate to population size and importation risks via land border crossings). We showed a modest number of strategically positioned ES sites can achieve a high early detection ability given assumed importation risks were geographically confined while dispersed importation risks reduced the effectiveness of ES. Our sensitivity analysis suggested that implementing the ES across large areas with low sampling frequency consistently resulted in a better early detection ability against various importation scenarios than implementing the ES in limited areas with high sampling frequency. Although we acknowledge the challenges of translating our simulated outcomes for real-world situations, our study has implications for deciding the scale and site selection of ES.

## Introduction

Global concerted efforts toward polio eradication have achieved a drastic reduction in the number of poliomyelitis cases [1] and cooperated surveillance systems

**Data availability statement:** All the data and code are deposited on GitHub (https://github.com/toshiakiasakura/polio_environmental_surveillance).

**Funding:** This study received the 2022-23 MSc Projects Funding (Trust Funds) from the London School of Hygiene & Tropical Medicine. TRA is supported by the Rotary Foundation (GG2350294), the Nagasaki University World-leading Innovative & Smart Education (WISE) Program of the Japanese Ministry of Education, Culture, Sports, Science and Technology and the Japan Society for the Promotion of Science (JSPS) KAKENHI Grant-in-Aid for JSPS Fellows (JP24KJ1827). KMO acknowledges funding from the Bill and Melinda Gates Foundation (INV-049314 and INV-049298). The funders had no role in study design, data collection and analysis, decision to publish, or preparation of the manuscript.

**Competing interests:** The authors have declared that no competing interests exist.

contributed to this achievement [2]. Patients with paralytic poliomyelitis are detected through syndromic surveillance, referred to as acute flaccid paralysis (AFP) surveillance, but a tiny portion of infections can be detected due to a very low paralysis-to-infection rate. It has been estimated that for every 200 wild poliovirus serotype 1 (WPV1) infections there will be one paralytic case [3]. To improve surveillance to rule out local transmission of poliovirus, sewage sampling, which is referred to as environmental surveillance (ES), has been developed. After the withdrawal of the trivalent oral polio vaccine (OPV) in 2016 and the emergence of vaccine-derived poliovirus serotype 2 (VDPV2) outbreaks [4,5], ES has been a vital complementary surveillance tool for polio eradication [6–8], especially for the detection of cryptic circulation in subnational areas of endemic countries, and detection of importation or confirmation of polio-free status in non-endemic countries.

Detection of poliovirus circulation through ES can trigger swift public health actions to contain outbreaks [9]. Recent examples include the detection of VDPV2 circulation first through the ES in the US [10,11] and the UK [12] in 2022, which enabled public health authorities to conduct active case finding, supplementary immunisation activity and social mobilisations. On the other hand, delays in detection have been linked to a large number of cases during outbreaks [13]. The long reporting delays of AFP surveillance have been attributable to a delay in sample collection, transport, culture and sequencing as well as the time required to ship collected samples to other countries due to the lack of facilities in resource-limited settings [13,14].

Expanding ES sites can enhance early detection capabilities, but the establishment and maintenance of ES sites incur costs and necessitate human resources [15]. To operate ES effectively, the quality assessment is essential, which comprises the appropriateness of sampling site locations [16], importation risk assessment [17], ES-covered population size, non-enterovirus detection [18], and the quality of sample handling and sample processing. Although guidelines for the implementation of ES have been developed by the Global Polio Eradication Initiative (GPEI) [19], specific guidance on the number and location of ES sites is still lacking due to uncertainties in available resources.

Quantitative evidence of the early detection ability of ES is needed to design ES layout strategies at the national or subnational level. One study empirically investigated the early detection capabilities using poliovirus genome data in Pakistan, showing that ES can detect the circulation of specific genotypic clusters before AFP surveillance in nearly 60% of sampled clusters [20]. From the perspective of a mathematical modelling approach, one seminal paper quantified the simulated cumulative probability of detecting poliovirus circulation through each AFP surveillance and ES [21], and one paper broadly examined the lead time of the first detection through ES over other surveillance systems assuming various pathogen characteristics [22]. Another study theoretically investigated optimal sampling frequency against emerging pathogens, considering a balance between sampling costs and disease burden [23].

There remains a gap in understanding the quantitative relationship between the ES early detection ability and spatial arrangements of ES sites. Given some flexibility in the selection of ES sites compared to laboratory facilities, we aim

to address how spatial arrangements of ES sites affect the early detection ability of ES using a simulation-based approach. Here, we employed the stochastic meta-population model to describe the geographical spread of WPV1 at 20 km resolution (i.e., 400 km$^2$) as well as the case-detection process through AFP surveillance and ES (following the basic model structure proposed by Ranta et al. [21]). Under varying ES spatial coverage, we quantified ES's early detection performance with detection patterns and the lead time of the first detection relative to AFP surveillance as primary outcomes. We further considered three importation risk distributions and two ES site layout strategies, totalling six scenarios, to account for uncertainties associated with importation routes and ES site layout. We utilised geographic and demographic characteristics of South Africa (where a polio-free status has been maintained since 1989 [24]) as a case study of a non-endemic country, which was motivated by the WPV1 importation event of 2022 into Mozambique, a neighbouring country to South Africa, implying a non-negligible risk of WPV1 importation into South Africa [25].

## Materials and methods

### Data

We collated population data for children under 5 years old in 100m spatial resolution in South Africa from WorldPop on 16$^{th}$ February 2024 [26]. We aggregated the dataset to a 20 km spatial resolution (i.e., 400 km$^2$), where we referred to each unit of area (matched with the real geographical location) as a 'patch' within a meta-population framework. We removed patches with less than 100 children under 5 years old for computational efficiency, resulting in 1502 patches in South Africa (S1 Fig in S1 text). We also collated population data for all age groups in South Africa, and both for children and all age groups in Mozambique to parameterise the population movement and ES sensitivity. National and district boundaries for South Africa and Mozambique were obtained from geoBoundaries [27] to visualise all the maps in this study. Furthermore, we used the district boundaries to relate district-level data to the corresponding patches for the meta-population model.

We collated the district-level vaccination coverage from the Expanded Programme on Immunisation National Coverage Survey Report 2020 in South Africa on 20$^{th}$ March 2024 [28]. South Africa's routine immunisation schedule includes both bivalent OPV and inactivated poliovirus vaccine (IPV). OPV is administered at birth and 6 weeks after birth, while IPV is administered as a part of the hexavalent vaccine (HEXA) at 6 weeks, 10 weeks, 14 weeks, and 18 months after birth. We calculated the proportion of children under 5 years old who were effectively immunised, considering the vaccine coverage for each dose and the vaccine effectiveness per dose, which we defined as the effective immunisation proportion (EIP) (S1 Table and S2 Fig in S1 text).

In addition, we obtained the wastewater plant information, including the location and their served-population size, from the National Institute for Communicable Diseases (NICD) on 18$^{th}$ March 2024. The detailed descriptions of the wastewater data can be found in S3-S4 Table and S8 Fig in S1 text. This is a simulation-based study for which no human data were used, and therefore no ethical approval is required.

### WPV1 transmission model

We constructed a stochastic meta-population model based on the SEIR compartment framework among unimmunised children under 5 years old considering the detection process of WPV1 through AFP surveillance and ES (S3 Fig in S1 text for a schematic representation). Briefly, a susceptible individual (S) enters the exposed state (E) when contacting infectious individuals. After the latent period (4 days), an exposed individual enters an infectious state (I), which lasts 15 days, then recovering from infection (R). We only considered unimmunised children under 5 years old since this age group contributes most to transmission in general, and those effectively immunised by routine vaccination were excluded from the dynamics. Additionally, we considered the process of births and deaths (i.e., removal from this age cohort) in the model.

Our model includes a geographical structure with its unit as a 'patch' and an infection hazard rate in each patch depends on the local dynamics (where homogenous mixing is assumed) as well as dynamics in other patches. The daily hazard rate for newly infected individuals in patch $i$ at day $t$ ($\lambda_{i,t}$) is expressed as

$$\lambda_{i,t} = \frac{\beta}{N_{i,c}} \left[ (1-\alpha) \left( I_{i,c,t} + I_{i,nc,t} \right) + \alpha \sum_{j \neq i} \pi_{ji} \left( I_{j,c,t} + I_{j,nc,t} \right) \right],$$

(1)

where $\beta$ corresponds to the transmission rate, $N_{i,c}$ corresponds to the population size of children under 5 years old of patch $i$ regardless of immunity status, $\alpha$ corresponds to the travelling rate (at which individuals are outgoing from the origin patch), and $\pi_{ji}$ denotes the rates of moving from origin $j$ to destination $i$. The moving rates ($\pi_{ji}$) were approximated by the radiation model, which showed a better fit to polio incidence than the gravity model [29–31]. To model the case-detection process through ES, we classified the infectious individuals into two classes though assuming no difference in infectiousness: infectious individuals at patch $i$ covered by the ES at day $t$ ($I_{i,c,t}$) and not covered by the ES ($I_{i,nc,t}$) with the probability of the patch-level ES population coverage ($p_c$, which is explained later) and $1 - p_c$, respectively.

We calculated $\beta$ from the relationship $R_0 = \beta/\gamma_2$ where $R_0$ corresponds to the basic reproduction number and $1/\gamma_2$ corresponds to an infectious period. We also define the effective reproduction number in the initial state at patch $i$ ($R_{e,i}$) as $R_0$ multiplied by EIP at patch $i$, which represents the reproduction number for imported cases. We assumed a relatively high $R_0$ of 14 compared to previous estimates [32,33]. This assumption resulted in 1.13 of the average $R_{e,i}$ across patches, and most $R_{e,i}$ centred around one with a maximum of around three (S9 Fig in S1 text). Additional details can be found in S1 text and parameters used in our model were summarised in S5 Table in S1 text.

## AFP and environmental surveillance model

Patients with WPV1 were assumed to develop AFP with a probability of 1/200 and assumed to seek healthcare on the same day as the paralysis onset. The incubation period of developing AFP was assumed to be 16.5 days [34,35] and we prepared six compartments with transition rates of 0.329 days$^{-1}$ to be aligned with the incubation period distribution [36]. We modelled the reporting of AFP cases as a binomial process with the probability considering the following three steps: patients visiting health care, being tested, and obtaining a positive result.

ES is assumed to be implemented in a fixed number of patches through each simulation and we varied the number of patches with ES (called as ES-covered patches) to assess the early detection ability of ES depending on spatial scale. Since observed ES population coverage in each district in South Africa ranged between 10.7% to 66.5% (S3 Table ), we introduced a patch-level ES population coverage ($p_c$) to represent incomplete ES catchment within ES-covered patches, meaning that only $p_c$% of individuals in ES-covered patches can enter the state $I_{i,c,t}$ in Equation 1. To choose the appropriate $p_c$ value, we further define a simulated national ES population coverage as the proportion of the population in the ES catchment area (i.e., those who can enter $I_{i,c,t}$ in Equation 1), which is obtained by the multiplication of $p_c$ and the proportion of ES-covered patches among all patches. We chose a default $p_c$ of 25% based on the observed ES location and the observed national ES population coverage (S3-S4 Table and S8 Fig in S1 text).

We modelled the case-detection process through ES as a binomial process. In the main analysis, we assumed the monthly wastewater sampling at ES-covered patches and the probability of detecting poliovirus from wastewater (called ES sensitivity) to be dependent on the density of infectious individuals. We employed the dose-response curve derived from COVID-19 incidence and wastewater samples in the US [37] to model this ES sensitivity (S6 Fig in S1 text), providing 50% probability of detecting poliovirus per sample given 2.3 infectious individuals per 100,000 population within a patch.

## Model implementation and outcomes

In each simulation, we randomly chose one patch to introduce WPV1 with a probability proportional to assumed importation risks of each patch. Then, we simulated a spread of WPV1 infection and a case-detection process through AFP

surveillance and ES until any of the following criteria were met: poliovirus circulation was detected through both surveillance systems; poliovirus was no longer circulated; or three years had passed from the simulation's start date.

The detection pattern for one simulation result falls into one of the following five categories: i) No detection, ii) AFP surveillance only detected polio patients, iii) AFP surveillance detected poliovirus circulation earlier than ES, iv) ES detected poliovirus circulation earlier than AFP surveillance, v) ES only detected poliovirus circulation (Fig 1B). We further classified categories iii) and iv) into two subcategories based on the 60-day cutoff for the lead time (LT) of the first detection (Fig 1C). We calculated a proportion of each category based on 10,000 simulations after we excluded the "No detection" category as a primary outcome since we are unable to confirm a failure of poliovirus detection for each importation in the real world (see S13 Fig in S1 text for results including "No detection" category). By iteratively varying the number of ES-covered sites and running 10,000 simulations for each, we determined and visualised the resulting proportions of detection patterns in a stacked area plot (Fig 1C). All the analysis was performed in Julia v1.8.3.

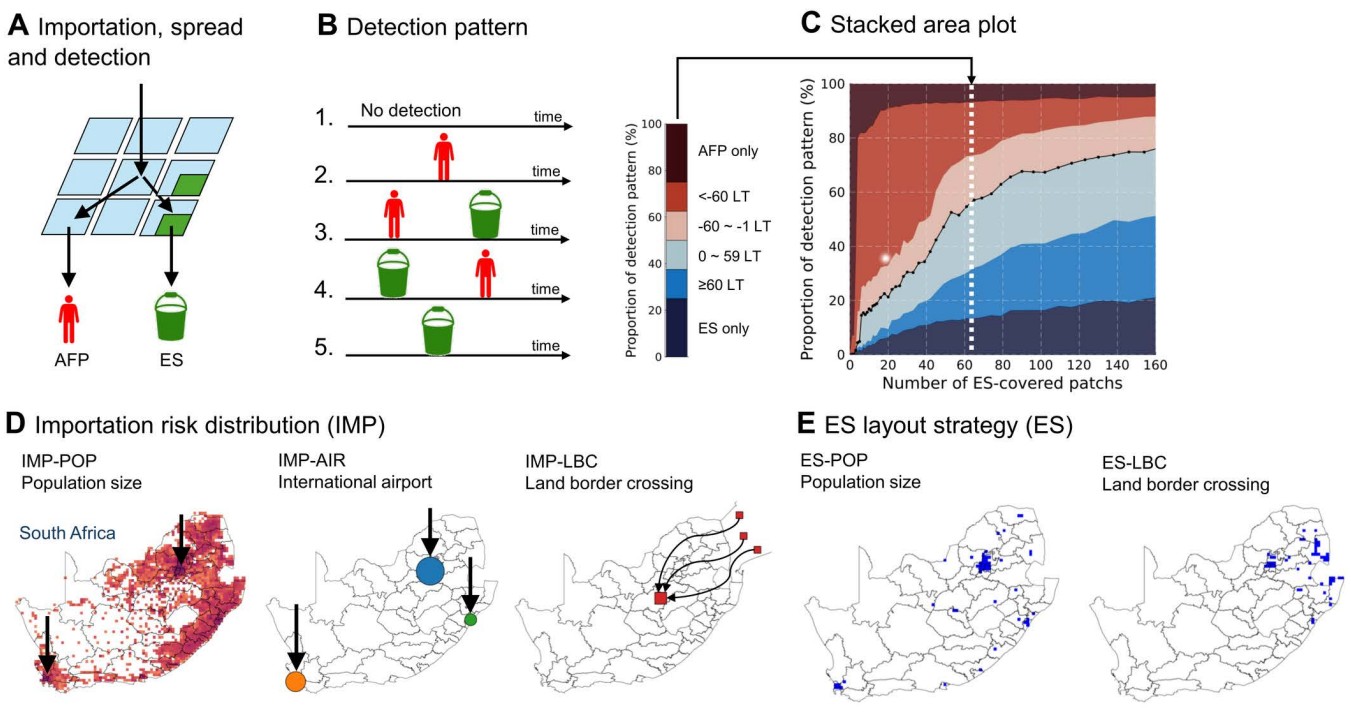

**Fig 1. Schematic illustration of study settings.** (A) Simulated importation of wild poliovirus serotype 1, spread described by the meta-population model, and detection through acute flaccid surveillance (AFP) operated in all patches and environmental surveillance (ES) operated in selected patches (in green). (B) Five detection patterns: i) No detection, ii) AFP only detection, iii) earlier detection through AFP surveillance than ES, iv) earlier detection through ES than AFP surveillance v) ES only detection. Detection patterns of iii) and iv) are further classified based on the lead time (LT) of the first detection through ES over AFP surveillance. (C) Stacked area plot of the proportion of each detection pattern against the number of ES-covered patches. (D) Illustration for the generating process of importation risk distributions: i) 'Population size'-based importation risk distribution (IMP-POP). Squares in orange-red colouration represent the population size of each patch in South Africa, which is assumed proportional to importation risks; ii) 'International airport'-based importation risk distribution (IMP-AIR). Each circle denotes the international airport in South Africa with a size proportional to the international inbound travel volume, which is associated with importation risks: Blue, O.R. Tambo International Airport; Orange, Cape Town International Airport; Green, King Shaka International Airport; iii) 'Land border crossing'-based importation risk distribution (IMP-LBC). Arrows illustrate population movement from multiple source patches in Mozambique to a single destination patch in South Africa, the sum of which is assumed proportional to importation risk for the destination patch in South Africa. (E) ES site layouts when the simulated national ES population coverage matched the observed one under i) 'Population size'-based ES site layout strategy (ES-POP) and ii) 'Land border crossing importation risk'-based ES site layout strategy (ES-LBC). Blue squares represent ES-covered patches. National and district borders for the maps were drawn with geoBoundaries under the CC-BY 4.0 license [27].

## Scenarios

We initiated each simulation by introducing WPV1 into a specific patch, but the first detection timing by ES is sensitive to our assumptions for a location of importation and a layout of ES-covered patches. To mitigate this, we opted to prepare various assumptions for importation risks and ES site layouts to evaluate the effect of spatial coverage of ES sites on the ES's early detection ability on a national scale.

We define an importation risk as the probability that WPV1 is introduced to a specific patch and an importation risk distribution as the distribution of importation risks across all patches in the country. Three importation risk distributions were prepared. 'Population size'-based importation risk distribution (denoted as IMP-POP) assumes the importation risk of each patch is proportional to the population size of each patch (Fig 1D, left). This distribution considers a movement of individuals importing poliovirus within a country until their first transmission. The other two importation risk distributions were weighted more on the importation route (e,g, via international air passengers or land-border crossings). 'International airport'-based importation risk distribution (IMP-AIR) assumes the importation risk of each patch is proportional to international inbound travel volume in 2019 and further considers mobilisation from airports (Fig 1D, middle). 'Land border crossing'-based importation risk distribution (IMP-LBC) considers an importation from Mozambique (as a case study considering the recent report of WPV1 detections [25]) and assumes the importation risk of each patch is proportional to travelling volume from Mozambique, which is approximated by the radiation model (Fig 1D, right).

We prepared two ES site layout strategies, which determine the order of patches to be covered by ES when we increase the number of ES-covered patches (S1-S2 Videos and Fig 1E). 'Population size'-based ES site layout strategy (denoted as ES-POP) assumes the ES is implemented in the descending order of population size of each patch. 'Land border crossing importation risk'-based ES site layout strategy (ES-LBC) assumes the ES is first implemented in a patch with a high importation risk via land border crossing from Mozambique. We included the land border crossing ES site layout strategy to assess the early detection ability against the poliovirus introduction in rural settings with an informed ES layout strategy (i.e., IMP-LBC/ES-LBC scenario). Representative layouts of ES sites were visualised for the two strategies, with simulated national ES population coverage matching 11.3% of the observed one. This results in 58 and 154 ES-covered patches for ES-POP and ES-LBC, respectively.

We considered the combination of each importation risk distribution and ES site layout strategy, totalling six scenarios: IMP-POP/ES-POP, IMP-AIR/ES-POP, IMP-LBC/ES-POP, IMP-POP/ES-LBC, IMP-AIR/ES-LBC, and IMP-LBC/ES-LBC. For example, IMP-AIR/ES-POP denotes the importation of WPV1 via international air passengers with an ES site layout prioritising areas with a large population size.

## Sensitivity analysis and weighted minimum distance to ES-covered patches

We performed the sensitivity analysis of $R_0$ (ranging from 8 to 16, corresponding to the average $R_{e,i}$ of 0.64 to 1.29), travelling rate between patches ($\alpha$), sampling frequency and ES sensitivity for the IMP-POP/ES-POP scenario. For a patch-level ES population coverage ($p_c$), sensitivity analysis was conducted for all six scenarios. Additionally, we conducted the sensitivity analysis under a single patch setting (i.e., simulations were performed without spatial structures) to differentiate between the effects of parameters on the ES's early detection ability attributable to spatial components and those stemming from model behaviours in a single patch (S10 Fig in S1 text).

Since our stochastic meta-population model requires a huge computational burden, we explored an alternative parsimonious assessment measurement for the ES's early detection ability. We considered the average minimum distance to each ES-covered patch from a patch with importation. If the distance is short, we expect a high ES's early detection ability. Since an importation location and the likeliness of growing outbreaks influence a detection process, we weighted the average minimum distance by importation risk in each patch and an outbreak probability. Here, we defined an outbreak probability as the probability of 10 or more infections occurring given $R_{e,i}$ under the branching process. The detailed descriptions can be found in S1 text.

## Results

### A modest number of targeted ES site implementations can achieve high simulated early detection probability

The proportion of each detection pattern was visualised as the stacked area plot against the number of ES-covered patches for six scenarios (Fig 2). We truncated the number of ES-covered patches at 160 sites even though the total number of patches for all of South Africa was 1502. The full scale of the figure can be found in S12 Fig in S1 text, in which the x-axis represents the national ES population coverage. We can expect a high early detection ability (i.e., more than 50%) from a small number of ES sites (i.e., covering 6–8 patches by ES) if importation risks were confined to limited geographical areas and ES was effectively implemented to cover those areas (e.g., IMP-AIR/ES-POP and IMP-LBC/ES-LBC, shown in Fig 2B and 2F). In comparison, if ES-covered areas were not matched with geographical areas with high importation risks (e.g., IMP-LBC/ES-POP and IMP-AIR/ES-LBC, shown in Fig 2C and 2E), we did not expect the 10–20 ES-covered patches to effectively detect poliovirus circulation earlier than AFP surveillance.

A high proportion of AFP only or ES only detection patterns in IMP-AIR indicate that patches with high importation risks in IMP-AIR are likely to have a high vaccination coverage, resulting in a small outbreak. This can be confirmed through the sensitivity analysis of $R_0$ within a single patch (S10 Fig in S1 text), showing low $R_0$ led to high AFP only or ES only detections, and the results including the "No detection" category (S12 Fig in S1 text), showing the highest proportion of "No detection" in IMP-AIR.

According to the wastewater plant data provided by the National Institute of Communicable Disease in South Africa, ES for poliovirus was operating at 17 wastewater plants, covering ~11.3% of the national population (if the wastewater-served

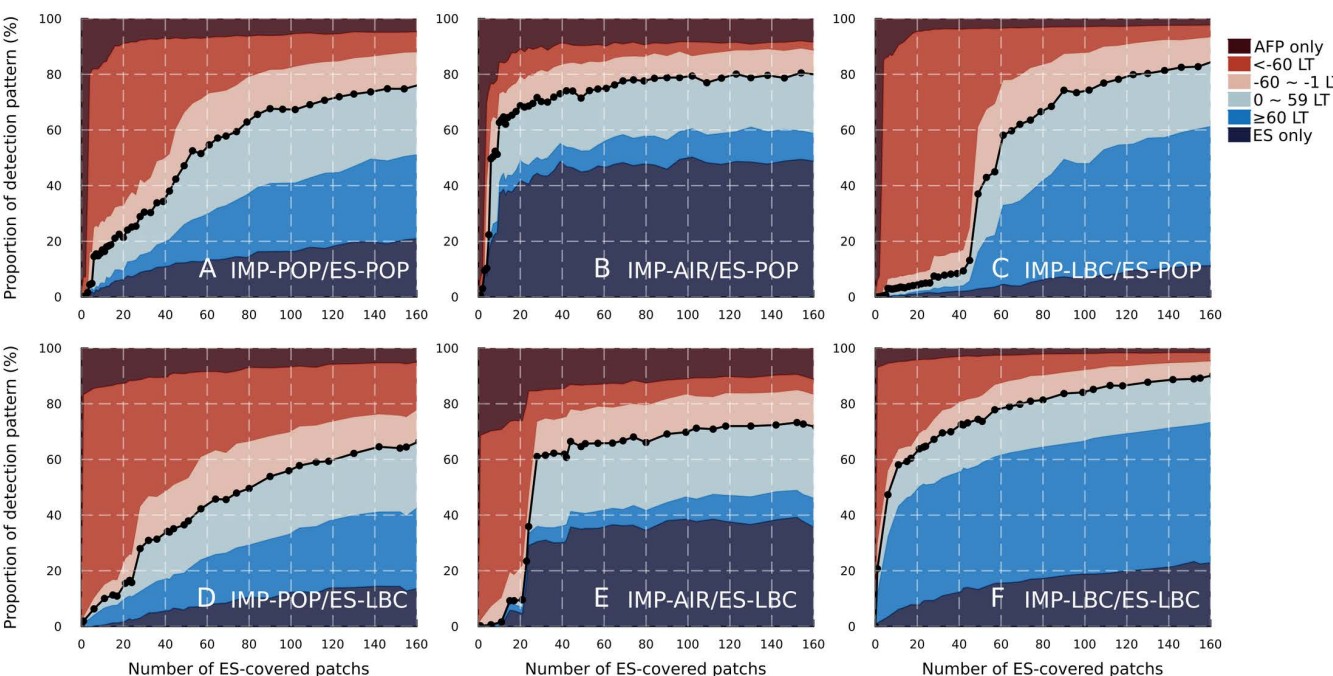

**Fig 2. Proportion of each detection pattern (%) against the number of ES-covered patches for six scenarios.** The blue-coloured area under the black dotted lines represents the simulated early detection probability, consisting of the early detection through ES over AFP surveillance and the ES only detection pattern. It is noted that the maximum number of ES-covered patches is 1502 but the x-axis is truncated at 160. IMP-POP, 'Population size'-based importation risk distribution; IMP-AIR, 'International airport'-based importation risk distribution; IMP-LBC, 'Land border crossing'-based importation risk distribution; ES-POP, 'Population size'-based ES site layout strategy; ES-LBC, 'Land border crossing importation risk'-based ES site layout strategy. LT denotes the lead time of the first poliovirus detection through ES over AFP surveillance.

population was imputed with the median value for two wastewater plants; see S1 text for details). Under our model assumptions, the closest number of ES-covered patches to achieve 11.3% of simulated ES national population coverage was 58 for ES-POP and 154 for ES-LBC.

### Sampling frequency and ES sensitivity are key parameters for enhancing early detection ability

We conducted a sensitivity analysis of the basic reproduction number ($R_0$), travelling rate between patches ($\alpha$), sampling frequency, ES sensitivity, and patch-level ES population coverage ($p_c$) for the IMP-POP/ES-POP scenario (Fig 3 and 4). The basic reproduction number did not influence the simulated early detection probability in the entire region of the number of ES-covered patches, whereas a lower $R_0$ resulted in a large proportion of AFP only or ES only detection patterns. This tendency was consistent with the sensitivity analysis under the single patch setting (S10 Fig in S1 text).

The simulated early detection probability remained consistent across different travelling rates ($\alpha$) for a small number of ES-covered patches (i.e., up to 40 patches). Even under a very high population movement assumption of $\alpha$ to be 0.50 (meaning 50% of individuals outgoing from one patch because of, for example, commuting to work or school), we observed only a 15% increase in simulated early detection probability after the number of ES-covered patches exceeded 50. Notable disparities were consistently observed when focusing on the proportion of AFP only or ES only detection patterns. Setting high travelling rates resulted in smaller proportions of ES only detection patterns whereas setting low travelling rates resulted in higher proportions.

Both sampling frequency and ES sensitivity (which is governed by two parameters of lognormal distribution) influenced the simulated early detection probability in the order of 10% to 25%, and these differences were consistently observed across the entire region of the number of ES-covered patches. It is noted that either with the higher sampling frequency or higher ES sensitivity, ES only detection pattern accounted for more than 30% among simulations excluding no detection pattern.

### Simulated early detection ability largely depends on the choice of patch-level ES population coverage ($p_c$)

We explored the impact of the patch-level ES population coverage ($p_c$) on simulated early detection ability for the IMP-POP/ES-POP scenario. The parameter $p_c$ modulated the balance between the concentration of ES placement in a single patch and the dispersion of ES sites across patches under a fixed national ES population coverage. The simulated early detection ability remained nearly consistent for $p_c$ values greater than 5%, given the same number of ES-covered patches (Fig 4A). Considering the observed district-level ES population coverages in South Africa all exceeding 10% (with a median of 22.5%), the early detection ability would be robust despite large variations in ES coverage in those districts (S3 Table in S1 text).

Since the higher $p_c$ value leads to a larger ES-covered population with limited ES-covered sites, we plotted the coloured points representing the simulation results where the simulated national ES population coverage matched the observed coverage (Fig 4A). Under such a constraint, when $p_c$ = 100%, only 5 sites were required to achieve the observed national ES population coverage but the combined sensitivity of these sites resulted in a low simulated probability of early detection (Fig 4A, green circle). Lower $p_c$ values correspond with many ES-covered patches, which in turn increased simulated early detection probability (Fig 4A, blue circle).

To further illustrate the role of the patch-level ES population coverage parameter, the same simulation results are displayed in two different x-axes. First, we employed the percentage of the population in ES-covered patches, which was calculated by the sum of the population in ES-covered patches divided by the total population size (Fig 4B). It is noted that (1- $p_c$)% of the population was not covered by ES but counted in the denominator. The tendency of simulated early detection probability was similar to Fig 4A. Second, we considered the national ES population coverage as the x-axis, which was calculated by multiplying $p_c$ with the "percentage of the population in ES-covered patches" to correctly account for

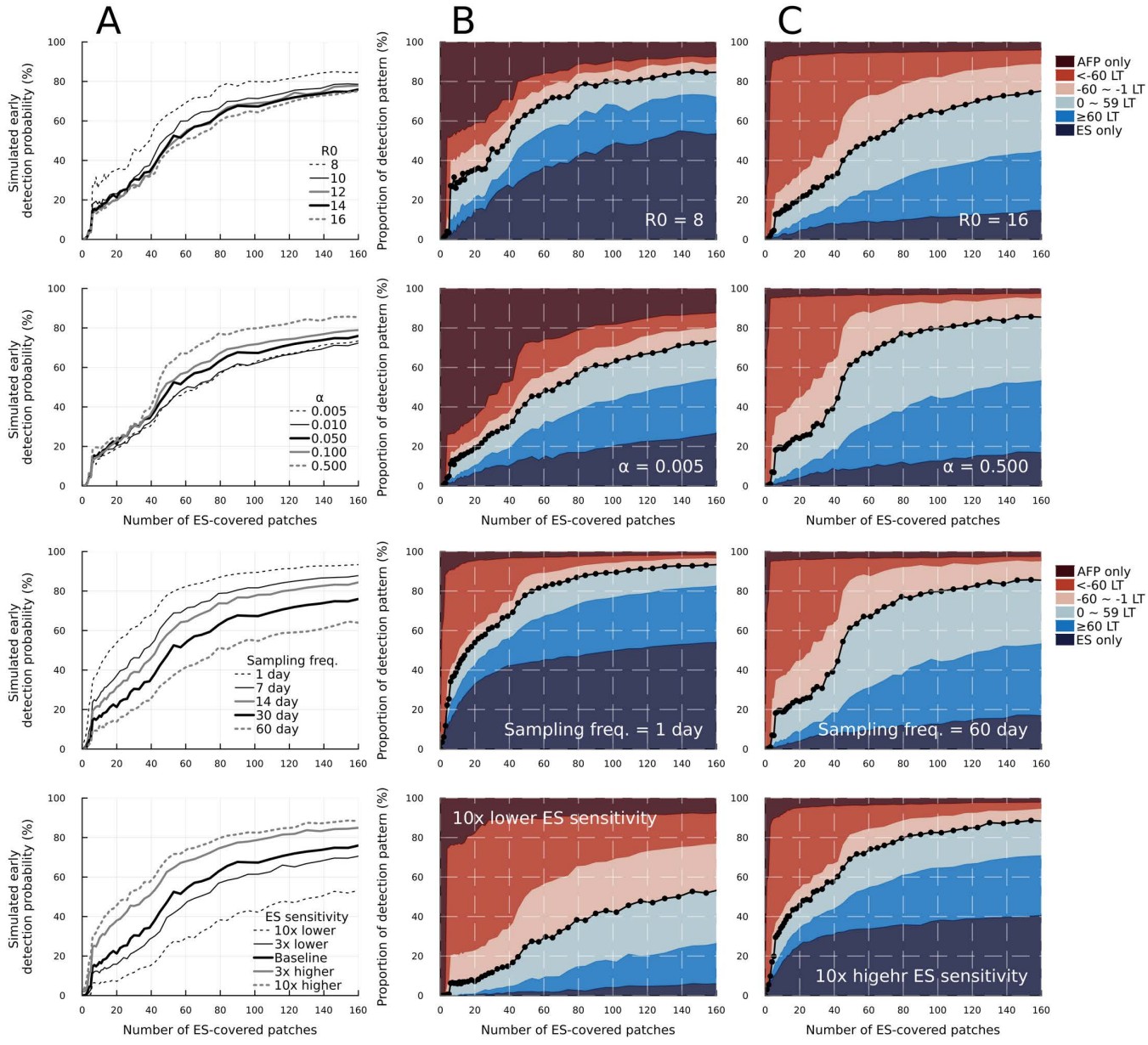

**Fig 3. Sensitivity analysis of the basic reproduction number (*R0*), travelling rate between patches (*α*), sampling frequency and ES sensitivity for the IMP-POP/ES-POP scenario.** (A) Simulated early detection probability for different parameters against the number of ES-covered patches. Bold black lines represent results with the same parameter value as in the main analysis. (B, C) Stacked area plot for the smallest and largest sensitivity parameter values. LT corresponds to the lead time of the first poliovirus detection through ES over AFP surveillance and 'sampling freq.' corresponds to the sampling frequency.

the population covered by ES. Even though the same national ES population coverage was maintained (Fig 4C, coloured points), the simulated early detection ability largely depended on the choice of $p_c$. This large variation implies the national ES population coverage would be an unreliable measurement to evaluate the spatial arrangements of the ES sites for the early detection ability. This tendency was consistent across the sensitivity analyses under the other five scenarios (S14-S15 Figs in S1 text).

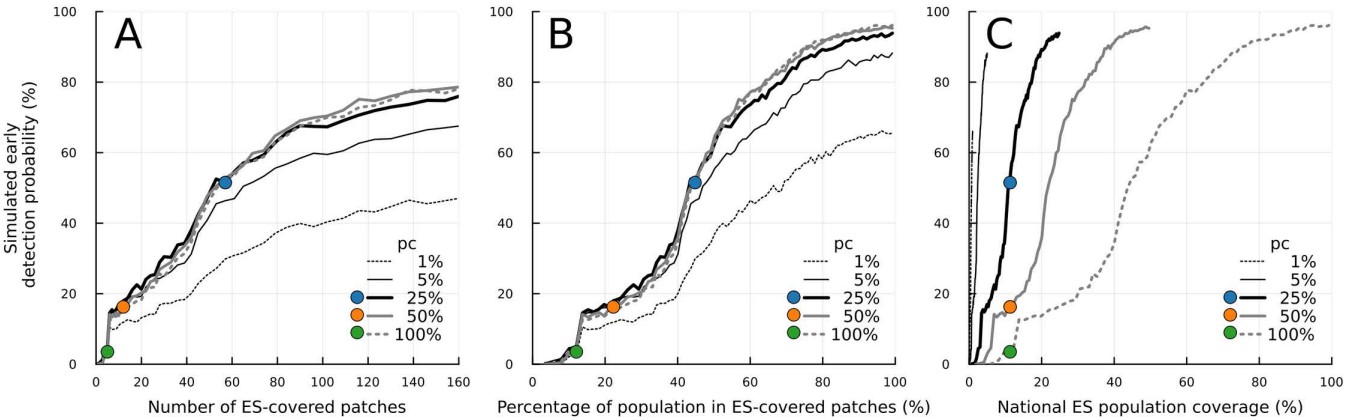

**Fig 4. Sensitivity analysis of the patch-level ES population coverage (*pc*) for IMP-POP/ES-POP scenario.** (A, B, C) Simulated early detection probability is plotted (A) against the number of ES-covered patches, (B) against the percentage of the population in ES-covered patches, and (C) the national ES population coverage. Black lines represent simulation results for each of $p_c$ and the number of ES-covered patches. The coloured data points in (A-C) represent simulations where the simulated national ES population coverage aligned with the observed national ES population coverage in South Africa (11.3%), under $p_c$ of 25% (blue), 50% (orange) and 100% (green). The national ES population coverage is given by the product of $p_c$ and the percentage of the population in ES-coverage patches. It is noted that the maximum number of ES-covered patches is 1502 and the x-axis for (A) is limited to a maximum value of 160.

### Correlations between simulated early detection probability and weighted average minimum distance to ES-covered patches

We explored more parsimonious assessment measurements for the ES site layout across the patches using the weighted average minimum distance to ES-covered patches. The relationship between simulated early detection probability and weighted average minimum distance showed a shape similar to an exponential curve for the IMP-POP and IMP-LBC scenarios (Fig 5A&B). Once the simulated early detection probability surpassed 50%, this relationship transitioned towards a nearly linear trend. Although the shape for these relationships was similar for ES-POP and ES-LBC scenarios, the scale of weighted average minimum distance was different.

In contrast, the relationship between simulated early detection ability and weighted average minimum distance showed an irregular pattern in IMP-AIR scenarios. We observed a sharp increase in simulated early detection probability despite a small difference in weighted average minimum distance in both ES-POP and ES-LBC. Conversely, only a small difference in simulated early detection probability was present despite a large difference in weighted average minimum distance. Notably, in the IMP-AIR/ES-LBC scenario, when the number of ES-covered patches exceeded 152, there was no increase in simulated early detection probability despite a more than 100 km decline in the weighted average minimum distance to ES-covered patches.

## Discussion

In this study, we assessed the quantitative relationship between early detection ability and the scale and locations of ES sites by employing a meta-population framework. We simulated the importation, spread and detection process, varying the number of ES-covered sites under different importation risk distributions and ES site layout strategies. By applying stochastic simulations, we successfully considered the partial detection patterns (i.e., ES only detection or AFP surveillance only detection) to align with real-world observations. The results provided here illustrated the potential of strategic positioning of ES sites to enhance early detection capabilities and clarified key ES-related parameters to be considered. Our results also highlight the importance of poliovirus importation risk assessment and how infectious disease surveillance should be tailored to perceived threats.

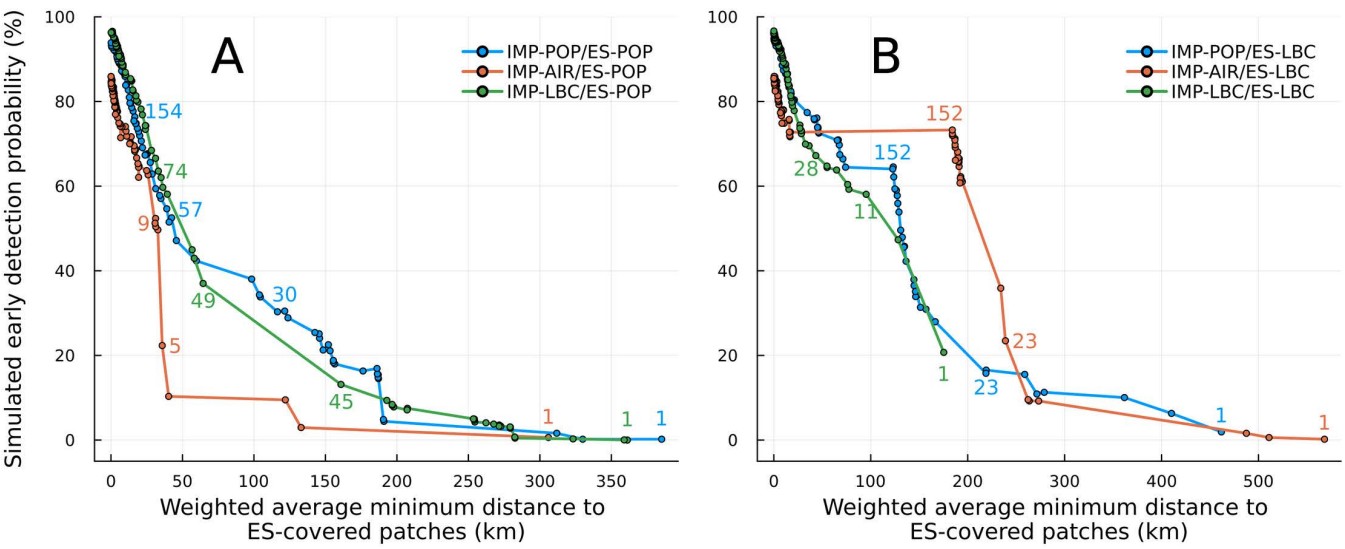

**Fig 5. Relationship between the weighted average minimum distance to ES-covered patches and simulated early detection probability (%).** (A) Under ES-POP scenarios. (B) Under ES-LBC scenarios. Minimum distances to ES-covered patches were weighted by importation risk and outbreak probability of 10 or more infections occurring, considering the effective immunisation proportion. The coloured numbers next to points correspond to the number of ES-covered patches for each importation risk distribution.

We found that simulated early detection probability exhibited a monotonic increase with the number of ES sites, but distinct variations in slope and plateau points were observed across six different scenarios. When importation risks were concentrated in confined patches, the modest number of strategic and targeted ES positioning can be highly efficient in the early detection of poliovirus circulation. Conversely, if importation risk was geographically dispersed, the effectiveness of ES was diminished, and many ES sites were required for a high early detection ability. It is noted that in our simulation study, we did not consider delays in reporting from the onset of AFP or delays in processing environmental samples. The reporting delays in patients with AFP were substantial in resource-limited settings, which could be around 29–74 days [22], and those delays should be considered in practice.

Our sensitivity analysis showed a different choice of the patch-level ES population coverage ($p_c$) resulted in large variations in simulated early detection probability even under the same national ES population coverage (Fig 4C), posing the challenge of translating our simulation results into the practical ES implementation strategy. This is because our current simplified patch assumption (i.e., 20 x 20 km grid) fails to accurately represent the complex geometry of the real-world wastewater catchment, lacking one-to-one spatial correspondence. Moreover, our model's implication that choosing as small $p_c$ as possible achieves the highest early detection ability is unrealistic, which is likely attributable to the violation of homogenous mixing assumptions within a patch for a small value of $p_c$. Considering those limitations, a current research gap for model assumptions is to identify areas where homogenous mixing is held, and in other words, to identify the fragmentation level of patches, as is historically pointed out by several authors [38–41]. One study of COVID-19 provided some insights into the homogenous mixing assumptions. This study used virus genome data with corresponding resident addresses in Dundee, Scotland, suggesting around a 5 km radius circle was well mixed in terms of genetic distance [42]. In addition, more granular data for wastewater catchment area and its population coverage can help develop a more accurate mathematical model, for example using a two-layer network model [43].

The sampling frequency is a parameter of interest to optimise the ES site layout [17,23]. By employing a daily sampling strategy in our study, the ES early detection ability increased significantly. However, this strategy resulted in an increased proportion of the ES only detection pattern, which might cause overacting against such an importation that does not lead

to secondary infections. Assuming an equal number of environmental samples can be taken for each month, extending ES sites can be more efficient in improving early detection ability and be robust against various importation risk distributions. For example, if we assumed 20 ES-covered sites and monthly sampling for the IMP-POP/ES-POP scenario, doubling sampling frequency led to a 10% increase in simulated early detection probability while doubling ES-covered sites resulted in the same increase and more resilience against multiple importation risk distributions. Moreover, the latter strategy can minimise the proportion of the ES only detection pattern. One concerning point is that expanding ES sites requires much more cost and additional human resources for transportation than increasing sampling frequency [16,44].

The basic reproduction number ($R_0$) for poliovirus is difficult to determine due to the small number of reported paralytic cases per outbreak and the impact of changes in hygiene. Our sensitivity analysis of $R_0$ showed small differences in simulated early detection probabilities, supporting the high robustness of results in the present study. Our assumed $R_0$ for the main analysis was based on a review of transmissibility by Fine et al. from 1999 [33], and the hygiene level in the African continent has since greatly improved. We therefore expect the current $R_0$ value in South Africa will be lower, but the sensitivity analysis suggested lower $R_0$ would not increase the probability of early detection but increased the ES or AFP surveillance only detection patterns. One study quantitatively investigated the early detection ability varying $R_0$ and other pathogen characteristics using a branching process [22] and found that the lead time for wastewater surveillance was different depending on $R_0$. Two reasons for differences between ours and that study can be considered. First, our model assumed an effective reproduction number of around one by considering vaccination coverage. So, the variation of the effective reproduction number was smaller compared to the range of $R_0$ investigated in Liu, et al. [22]. Second, we considered ES only detection patterns as early detection, which did not happen in the branching process model.

Our study is not free from limitations. First, we considered the country where OPV and IPV are routinely administered. The large outbreak of VDPV2 was caused by switching from trivalent OPV to bivalent OPV [45], and many countries are planning to cease OPV usage except for outbreak response [46,47]. Moreover, in many developed countries, IPV is only included in their routine immunisation schedule. Those vaccinated with only IPV could spread poliovirus to others due to lack of mucosal immunity and could be detected through ES, but they would be less likely to develop AFP due to the humoral immunity, implying increased utility of the ES compared to our study findings (where an IPV-OPV schedule is assumed). Second, we only focused on WPV1 and did not consider other serotypes (such as cVDPVs) or transmission from immune-compromised individuals shedding (iVDPVs).

Third, we limited transmission dynamics in children under 5 years old, assuming those aged 5 or more were completely immunised. However, the reported age distribution of poliomyelitis cases is skewed towards older groups in non-endemic countries [48], and multiple reasons for being unvaccinated were considered such as migration, poverty and conflict [49–51]. We ignored those pocket populations considering the size of that population and the paucity of historical vaccine coverage data in South Africa. Fourth, our importation risk distributions were essentially based on the population size of each patch and the approximated mobilisation pattern by the radiation model, which could make the ES performance better when compared to reality. Quantitative data about international traveller movement from airports and border crossing populations could improve predictions that support risk assessment. Furthermore, research on the importation pathway is demanding for prevention, detection and response. The importation routes to reported outbreak sites were often unknown and one to three years of cryptic circulation was suspected for some outbreaks [25]. Lastly, parameters for ES sensitivity, which was described by the lognormal distribution, were estimated using the wastewater surveillance data for COVID-19 in the US [37]. Considering differences in virus shedding characteristics between poliovirus [52] and COVID-19 [53,54] as well as differences in ES site system and quality between the US and other low- and middle-income countries, the ES sensitivity will be inherently different, underscoring the need for region-specific data to assess site-specific ES sensitivity.

In conclusion, several countries are planning to initiate, expand and optimise the ES for poliovirus. We varied the number and location of the ES sites under different importation risk distributions to quantify the ES early detection ability

over AFP surveillance. Our results showed that risk-targeted ES site layout could achieve high early detection capabilities. Further research is required to optimise resources for the ES to monitor the progress toward polio eradication.

## Supporting information

**S1 Text. Supplementary methods and results.**
(PDF)

**S1 Video. Simulated incremental implementation of environmental surveillance (ES) with the 'Population size'-based ES site layout strategy (ES-POP).** ES sites were implemented in descending order of the population size of each patch. Blue squared areas represent patches covered by ES sites.
(GIF)

**S2 Video. Simulated incremental implementation of environmental surveillance (ES) with the 'Land border crossing importation risk'-based ES site layout strategy (ES-LBC).** ES sites were first implemented in a patch with a high importation risk via land border crossing from Mozambique. Blue squared areas represent patches covered by ES sites.
(GIF)

## Acknowledgments

We thank Juliet R. C. Pulliam for insightful comments to improve result presentations, Kerrigan McCarthy for sharing wastewater plant-related information and Emily Nightingale for valuable comments to improve manuscripts.

## Author contributions

**Conceptualization:** Toshiaki R. Asakura, Kathleen M. O'Reilly.

**Data curation:** Toshiaki R. Asakura.

**Formal analysis:** Toshiaki R. Asakura.

**Funding acquisition:** Toshiaki R. Asakura, Kathleen M. O'Reilly.

**Investigation:** Toshiaki R. Asakura.

**Methodology:** Toshiaki R. Asakura, Kathleen M. O'Reilly.

**Resources:** Toshiaki R. Asakura, Kathleen M. O'Reilly.

**Software:** Toshiaki R. Asakura.

**Supervision:** Kathleen M. O'Reilly.

**Validation:** Toshiaki R. Asakura.

**Visualization:** Toshiaki R. Asakura.

**Writing – original draft:** Toshiaki R. Asakura.

**Writing – review & editing:** Kathleen M. O'Reilly.

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
