## [Decision Letter · Decision Letter 0]

Dear Dr. Asakura,

Thank you for submitting your manuscript to PLOS ONE. After careful consideration, we feel that it has merit but does not fully meet PLOS ONE’s publication criteria as it currently stands. Therefore, we invite you to submit a revised version of the manuscript that addresses the points raised during the review process.

**ACADEMIC EDITOR: Please insert comments here and delete this placeholder text when finished.**

This manuscript has a potential to contribute to the available literature on environmental surveillance.The manuscript could be better structured using the conventional (1. Introduction 2. Methods 3. Results and 4. Discussion) format as against the current format of the manuscript (which has 1. Introduction 2. Results 3. Discussion and 4. Methods). This may also help address the concern around repetition of the methods in the results section noticed.Some concerns have been raised around the assumptions made by the authors in the manuscript and how some of the limitations are described. Please address any of these concerns.While the authors have listed and highlighted three limitations, it may be beneficial to also include why and how these limitations do not invalidate the results of the study.The manuscript is well written, and results fairly well discussed with clear conclusions. Some concerns around the modelling assumptions and also results and figures in the results will benefit from revision to enhance clarity and better understanding of the results.While this manuscript focuses on results from modeling, how does the outcome of this address real world situations may be of particular public health interest. It may be nice to address this is in the revision.

publication criteria  and not, for example, on novelty or perceived impact.

We look forward to receiving your revised manuscript.

Kind regards,

Terna Ignatius Nomhwange, MD,DTM&H,MBA

Academic Editor

PLOS ONE

Journal Requirements:

2. Thank you for submitting your work to PLOS ONE. At this time, we require the following information in order to proceed with your submission: As you are reporting a retrospective study of medical records or archived samples, please ensure that you have discussed whether all data were fully anonymized before you accessed them and/or whether the IRB or ethics committee waived the requirement for informed consent. If patients provided informed written consent to have data from their medical records used in research, please include this information.

3. We note that Figures 1, S1 and S2 in your submission contain [map/satellite] images which may be copyrighted. All PLOS content is published under the Creative Commons Attribution License (CC BY 4.0), which means that the manuscript, images, and Supporting Information files will be freely available online, and any third party is permitted to access, download, copy, distribute, and use these materials in any way, even commercially, with proper attribution. For these reasons, we cannot publish previously copyrighted maps or satellite images created using proprietary data, such as Google software (Google Maps, Street View, and Earth). For more information, see our copyright guidelines: http://journals.plos.org/plosone/s/licenses-and-copyright.

a. You may seek permission from the original copyright holder of Figures 1, S1 and S2 to publish the content specifically under the CC BY 4.0 license.  

Additional Editor Comments:

Thank you for the submission of your manuscript highlighting the role of spatial arrangements in environmental surveillance for Polio.

I am sharing some suggestions and comments made by reviewers to help strengthen the quality of the manuscript.

Please feel free to update based on these reviews and provide any justifications for any suggestions you decide not to accept.

Reviewers' comments:

Reviewer's Responses to Questions

**Comments to the Author**

1. Is the manuscript technically sound, and do the data support the conclusions?

Reviewer #1: Yes

Reviewer #2: Yes

Reviewer #3: Yes

2. Has the statistical analysis been performed appropriately and rigorously?

Reviewer #1: Yes

Reviewer #2: Yes

Reviewer #3: Yes

3. Have the authors made all data underlying the findings in their manuscript fully available?

Reviewer #1: Yes

Reviewer #2: Yes

Reviewer #3: Yes

4. Is the manuscript presented in an intelligible fashion and written in standard English?

Reviewer #1: Yes

Reviewer #2: Yes

Reviewer #3: Yes

Reviewer #1: Summary: This interesting study used simulation modeling to highlight and explore the potential of early detection of polio outbreaks by environmental surveillance (ES). The authors demonstrated how the geography of importation affects detection probability and the utility of ES as an early detection tool. The introduction is strong, and the conclusions are sensible and useful. I have some suggestions that may improve the clarity of the methods and results for the reader. I think the paper will be a useful addition to the ES literature.

Comments

1. I think that this paper suffers from trying to report the results before the methods, because the authors end up having to say the methods twice, and incompletely/confusingly the first time. I recommend reordering the paper with the methods first, so that the results can be more easily understood.

2. Figure 1. I had trouble processing the image associated with IMP-LBC. First, it wasn’t immediately clear visually that you were showing South Africa and Mozambique. Second, I think that the intention is to highlight transmission risk along the South Africa-Mozambique border, but it seems that only one of the three arrows is across the border. Perhaps you could show South Africa the same size as in the rest of the bottom row of figures, with the border of Mozambique outlined and with arrows crossing the border?

3. Figure 1 C and Figure 2. In Figure 1B you describe “no detection” as an option, but it’s not included in these results figures. Was “no detection” not an outcome that was observed? Ah, later you say that it was an outcome but that it’s included in the supplementary material. It’s possible that I missed it, but, as a reader, I could have used some up front notification and justification of why the “no detection” was not being included. Was it because the importation scenario is non-modifiable? Yes, the more that I think about it, the more that I think it would be helpful for you to first talk about (or show) the “no detection” vs “any detection” by scenario before getting into the lead-time results.

4. Figure 2. The scenario abbreviations are cumbersome. Perhaps writing titles for the subpanels would be clearer.

5. Interestingly, compared to the IMP-POP scenarios, there was a greater proportion of both AFP only and ES only detection in the scenarios. Can you say more about why that’s happening? (Probably fewer cases means harder to detect through any means?) It comes up again with the “low R0” sensitivity analysis, so it’s worth orienting the reader to.

6. The R0 values (10-18) seem high to me. I normally think of polio as being in the 5-7 range, though I’m not an expert on polio specifically. Additionally, you haven’t described in the Methods how R0 is connected to the parameterization of the model (is changing R0 just changing beta?), nor have you described/justified the spread of the values of the parameters in the sensitivity analyses performed.

7. The values of alpha in the Results are fairly meaningless to the reader. What does it mean, practically, to go from alpha= 0.05 to 0.5?

Reviewer #2: The paper “Assessing early detection ability through spatial arrangements in environmental surveillance for poliovirus: a simulation-based study” is a well written and technically sound paper. Below are some comments to the authors:

1. In Fig 1. (D), the IMP-LBC map seems a bit different than the others (I am guessing the authors are showing a subarea of the whole map). Can you please explain which region the map refers to? It will be nice to have a similar map to the rest for easy understanding.

2. It seems natural that ES strategy that aligns well with the IMP route will perform better. For instance, the ES-POP strategy covers the IMP-AIR pretty well, as the airports are located in high-density locations, hence the result is supposed to be good. In other words, IMP-Air can be considered as a subset of the IMP-POP. Can you justify the inclusion of this scenario IMP-Air? Also, the authors’ contribution beyond this natural alignment is not clear.

3. Lines 182-185, it’s hard to interpret whether the result is good or bad. More clarification will be helpful.

4. The authors mentioned expanding the ES sites that comes with related maintenance costs. But they haven’t provided any cost-benefit analysis, for instance, if 54 ES-POP sites are enough to achieve a 11.3% national ES population coverage, will this implementation be cheaper than the alternatives or financially doable?

5. The sensitivity analysis was conducted for the IMP-POP/ES-POP scenario only. Can you please clarify why this one is chosen?

6. Lines 209-211, the sensitivity analysis results seem natural, as if people travel frequently and more patches are used, the early detection rate will be higher. A more rigorous sensitivity analysis can be helpful to understand the effectiveness of ES, for example, using interaction of the factors (such as basic reproduction and travel rate) against the number of patches.

7. Lines 497-498, if n_{i,t} is an index variable, then how can it be used to characterize the Binomial distribution(n, p), where n = the total number of samples takes. Please, clarify.

Reviewer #3: The manuscript demonstrates scientific merit but would benefit from improvements in structural organization, refinement of model assumptions regarding immunized individuals, and stronger efforts to link simulation patches to real-world geography.

The abstract is 276 words, which appears to be relatively long compared to typical journal standards. Consider reducing the length of the abstract to enhance conciseness and better align with common practices in academic journals.

The introduction is well-written, demonstrating fluency and a clear articulation of the research problem, motivation, and support from existing literature.

The structure is unconventional, with the Results section preceding the Materials and Methods section. While this structure is not strictly prohibited by PLOS ONE, it may reduce the clarity and logical flow of the manuscript.

Authors assume that transmission is limited to children under 5, with individuals aged 5 and above being fully immunized. This overlooks the potential role of immunized individuals as asymptomatic carriers and vehicles for poliovirus transmission. Dont you think?

The authors note that “it is difficult to relate a patch in our simulations to real-world geographical settings.” However, while this is acknowledged, it is plausible to relate patches to administrative units, population density maps, or urban-rural divisions.

Results are robust, but the authors acknowledge challenges in translating simulated outcomes to real-world settings. Continue emphasizing the limitations and uncertainties associated with model-based simulations, while discussing avenues for future validation and refinement using empirical data.

**Do you want your identity to be public for this peer review?** For information about this choice, including consent withdrawal, please see our Privacy Policy

Reviewer #1: No

Reviewer #2: No

Reviewer #3: No

---

## [Author Response · Author response to Decision Letter 1]

9 Apr 2025

We have uploaded "Response to Reviewers.pdf".

---

## [Decision Letter · Decision Letter 1]

Assessing early detection ability through spatial arrangements in environmental surveillance for poliovirus: a simulation-based study

PONE-D-24-46215R1

Dear Dr. Asakura,

We’re pleased to inform you that your manuscript has been judged scientifically suitable for publication and will be formally accepted for publication once it meets all outstanding technical requirements.

Kind regards,

Terna Ignatius Nomhwange, MD,DTM&H,MBA

Academic Editor

PLOS ONE

Additional Editor Comments (optional):

I believe the authors have addressed all the concerns previously raised by the reviewers and have  submitted a  stronger version of the manuscript. There are a few editing corrections which I believe will be addressed as part of the publication process.

Thank you.

Reviewers' comments:

Reviewer's Responses to Questions

**Comments to the Author**

Reviewer #1: (No Response)

Reviewer #2: All comments have been addressed

2. Is the manuscript technically sound, and do the data support the conclusions?

Reviewer #1: Yes

Reviewer #2: Yes

3. Has the statistical analysis been performed appropriately and rigorously?

Reviewer #1: Yes

Reviewer #2: Yes

4. Have the authors made all data underlying the findings in their manuscript fully available?

Reviewer #1: Yes

Reviewer #2: Yes

5. Is the manuscript presented in an intelligible fashion and written in standard English?

Reviewer #1: Yes

Reviewer #2: Yes

Reviewer #1: My comments are adequately addressed. I have a couple of minor suggested corrections to the descriptions of some of the model parameters.

Ln 143: “α corresponds to the travelling rate (representing a proportion of individuals outgoing from one patch).” A quantity cannot be both a rate and a proportion. I believe you want to say something like, “α represents the proportion of individuals traveling out of the patch at any given time.”

Ln 152: “1/γ corresponds to a latent period.” I believe that you mean “1/γ corresponds to the infectious period.” Unless I'm missing it, I think you haven't provided values for your infectious period or latent period in the text. I also think you should use consistent labeling of your parameters between the main text and supplement (γ in the main text is, presumably, the rate of leaving the infectious period, but it's γ2 in the supplement).

Table S5: There are multiple incorrect descriptions of the parameters conflating rates, periods, and fractions. Revisit α, γ1, and γ2 in particular, but recommend a thorough read-through to give accurate descriptions.

Reviewer #2: (No Response)

**Do you want your identity to be public for this peer review?** For information about this choice, including consent withdrawal, please see our Privacy Policy

Reviewer #1: No

Reviewer #2: No

---

## [Editor Report · Acceptance letter]

PONE-D-24-46215R1

PLOS ONE

Dear Dr. Asakura,

I'm pleased to inform you that your manuscript has been deemed suitable for publication in PLOS ONE. Congratulations! Your manuscript is now being handed over to our production team.

Kind regards,

on behalf of

Dr. Terna Ignatius Nomhwange

Academic Editor

PLOS ONE